# Discovery of Transfer Factors in Plant-Derived Proteins and an In Vitro Assessment of Their Immunological Activities

**DOI:** 10.3390/molecules28247961

**Published:** 2023-12-05

**Authors:** Mesfin Yimam, Teresa Horm, Shengxin Cai, Alexandria O’Neal, Ping Jiao, Mei Hong, Thida Tea, Qi Jia

**Affiliations:** 1Unigen Inc., 2121 South State Street, Suite #400, Tacoma, WA 98405, USA; scai@unigen.net (S.C.); pjiao@unigen.net (P.J.); meih@unigen.net (M.H.); ttea@unigen.net (T.T.); qjia@unigen.net (Q.J.); 2Department of Biology, Pacific Lutheran University, 12180 Park Ave. S, Tacoma, WA 98447, USA; 3Seagen Pfizer, 21717 30th Dr SE, Bothell, WA 98021, USA; alexhoneal@gmail.com

**Keywords:** transfer factor, natural killer cells, plant proteins, immune support, *Brassica juncea*, *Raphanus sativus*

## Abstract

Repeated exposure to pathogens leads to evolutionary selection of adaptive traits. Many species transfer immunological memory to their offspring to counteract future immune challenges. Transfer factors such as those found in the colostrum are among the many mechanisms where transfer of immunologic memory from one generation to the next can be achieved for an enhanced immune response. Here, a library of 100 plants with high protein contents was screened to find plant-based proteins that behave like a transfer factor moiety to boost human immunity. Aqueous extracts from candidate plants were tested in a human peripheral blood mononuclear cell (PBMC) cytotoxicity assay using human cancerous lymphoblast cells—with K562 cells as a target and natural killer cells as an effector. Plant extracts that caused PBMCs to exhibit enhanced killing beyond the capability of the colostrum-based transfer factor were considered hits. Primary screening yielded an 11% hit rate. The protein contents of these hits were tested via a Bradford assay and Coomassie-stained SDS-PAGE, where three extracts were confirmed to have high protein contents. Plants with high protein contents underwent C18 column fractionation using methanol gradients followed by membrane ultrafiltration to isolate protein fractions with molecular weights of <3 kDa, 3–30 kDa, and >30 kDa. It was found that the 3–30 kDa and >30 kDa fractions had high activity in the PBMC cytotoxicity assay. The 3–30 kDa ultrafiltrates from the top two hits, seeds from *Raphanus sativus* and *Brassica juncea*, were then selected for protein identification by mass spectrometry. The majority of the proteins in the fractions were found to be seed storage proteins, with a low abundance of proteins involved in plant defense and stress response. These findings suggest that *Raphanus sativus* or *Brassica juncea* extracts could be considered for further characterization and immune functional exploration with a possibility of supplemental use to bolster recipients’ immune response.

## 1. Introduction

Repeated exposure to environmental stress (such as pathogens) leads to the evolutionary selection of adaptive traits. Many species transfer immunological memory to their offspring to meet immune challenges. In mammals, including humans, known to possess adaptive immune systems, intergenerational protection is achieved by the transfer of antibodies through the placenta or breast milk; other vertebrates, such as birds and fish, transfer antibodies to their offspring via their eggs. Plants and invertebrates that lack adaptive immunity pass their defense experience to their progeny through their DNA by boosting the non-specific defenses that make up their offspring’s innate immune system with the advantage of lifelong generational protection [1].

The transfer factor is among the many mechanisms in humans where transfer of immunological memory can be achieved for enhanced immunological responses of a recipient. Multiple hypotheses have been proposed regarding the nature of the transfer factor. While some suggest that the transfer factor is one or more peptide(s) with the capacity to produce an immediate and/or delayed immunological response with a molecules mass range of >3500 and <12,000 Da, others suggest that the transfer factor is a single substance that achieves diverse responses by acting on various cellular receptors [2]. Likewise, multiple mechanisms of action (MOAs) have been proposed for the transfer factor. Perhaps the most consistent MOA of the transfer factor is at the level of cell-mediated immunity, where the transfer factor could transmit the ability to express delayed hypersensitivity reactions and cell-mediated immunity from an individual or donor previously sensitized to an antigen to a non-immune or non-sensitized receptor. This knowledge has advanced its usage for prevention of infections (viral, parasitosis, fungal and mycobacterial), primary immunodeficiencies (PIDs), atopy and cancer [3]. While the main effect of transfer factors on the immune system is reported as expression of delayed-type hypersensitivity, there still is a possibility that the action is nonspecific, which could produce a generalized adjuvant effect, leading to an amplified immune response [4]. In fact, immunologically active peptides have been identified from non-antigen-specific human dialyzed lymphocyte extracts produced at the National School of Biological Sciences (IPN, National Polytechnic Institute) [5], suggesting that individuals whose immune systems are intact could gain the benefit of the transfer-factor-like moiety effect to bolster and strengthen their immune system.

Besides human blood with antigen specificity, other sources of transfer factors such as bovine leukocytes, bovine colostrum and egg yolks have been reported [6,7,8]. Nonantigen-specific transfer factors from bovine colostrum and egg yolk sources have been used as oral supplements in the market for nonspecific, generalized innate immune support in humans. Despite its significant beneficial effect, the widespread application of the transfer factor is limited due to its scarce animal origin, sourcing and sustainability. These factors warrant the search for an alternative from natural plant sources. In the current report, plant protein/peptides were considered as an alternative source and screened for their efficacy.

Plants have a complex immune system to recognize and protect themselves against pathogens. Plants lack circulating defender cells and an adaptive immune system [9]. Instead, they have evolved their innate immune system to recognize damage-associated molecular patterns and pathogen-associated molecular patterns via pattern-recognizing receptors to launch diverse pattern-triggered immunity (PTI) [10]. Plant peptides play a significant role in plant defense. Plant defense peptides are believed to be proteins with a length of <100 amino acids [11]. Well-characterized, biologically active plant peptides could have beneficial effects in humans by enhancing the innate immune response for a robust protection against daily immune challenges.

Small molecules such as brassinosteroids [12], gibberellins [12], kinetin [13] and peptide-like systemins [14] play important roles in plant innate immunity. In this study, we sought to find plant-based proteins/peptides that behave like a transfer factor to boost human innate immunity. Based on the prior reports of defense peptides in plant immunity [10,15,16,17,18] and an in-house high-protein plant study, 4331 plants in the Unigen Phytologix library were screened and a total of 100 plants were selected for this study to isolate those with high protein contents and prior use as immune boosters.

The use of cryopreserved human PBMCs in functional and phenotypic immunological assays has granted significant understanding about the role of NK cell functions. Natural killer (NK) cells are potent effectors of the innate immune system and form the first line of defense. They are members of the innate immune cell family and are characterized in humans by expression of the phenotypic marker CD56 in the absence of CD3. NK cells are the most abundant innate cytotoxic lymphoid in humans with an immediate cytotoxic effect. NK cells do not require prior antigen priming for their activity; instead, their activation is initiated by various receptors with activating or inhibitory functions. NK cells produce diverse cytokines, growth factors and chemokines that could shape the immune response of a host by interacting with dendritic cells, macrophages and T cells [19]. As NK cells constitute 5–30% of PBMCs [20,21], in our primary screening we used a PBMC cytotoxicity assay to assess the NK cytotoxic effect against human cancerous lymphoblast cells—K562 cells after incubation with plant peptides through presumed increased activity of NK cells as an effector without the need for purified NK cells.

## 2. Results

### 2.1. Primary Screening

PBMCs were incubated overnight with 50 and 100 μg/mL of aqueous extracts (AEs) of the 100 selected plants in duplicate, with water as a vehicle. IL-2 (20 μg/mL) and colostrum-based TF (C-TF hereafter) (500 μg/mL) were used as positive controls. K562s were added the next day in a ratio of 25:1 effector/target cells, and a K562 well with no PBMCs was also used as a control. This ratio was selected based on data collected at the time of method optimization. The percent cytotoxicity (measure of effectiveness) of each sample was normalized to the negative control, which had PBMCs and K562s incubated together without pre-treatment, and to the IL-2 positive control, which was set to 100% cytotoxicity.

Eleven plant aqueous extracts (AEs) showed significant efficacy in K562 cell killing, resulting in an 11% hit rate (Table 1) with the percentage cell killing at 100 μg/mL higher than colostrum as a positive control at 500 μg/mL. Effectiveness ranges of 42.9–77.9% and 55.3–75.2% were observed for these plant extracts at 50 and 100 μg/mL concentrations, respectively. These effects were dose-correlated for P00303, P00397, P00447, P00703, P09153, P09495 and R00813. While there was no difference in inhibition of growth between the two concentrations for R00137 and R00659, a slight increased inhibition in growth at 50 μg/mL compared to 100 μg/mL was observed for R00918 and P08435 (Figure 1).

### 2.2. Protein Content as Measured by a Bradford Assay

The eleven primary hits from the screening were tested in a Bradford assay for their protein content as described in the methods. Each aqueous plant extract contained measurable protein (Figure 2). The highest protein content, 688.4 µg/mL, was found in aqueous extract R00137. The lowest protein content, 43.4 µg/mL, was found in aqueous extract R00659. The C-TF showed 1301.5 µg/mL of total proteins.

### 2.3. Protein Content on SDS-PAGE Gel

The protein content of the AE extracts of the 11 primary hits was further confirmed on SDS-PAGE gels stained with Coomassie blue (Figure 3). Here, again, extract R00137 showed a significant protein content, followed by P00303 and P00397. The majority of protein in these extracts seemed to be less than 30 kDa. At least in this method, the protein content for the other extracts was very minimal.

### 2.4. Methanol Fractions of Top Three Hits on SDS Page

C18 column fractions, as described in the methods, for the top three protein-rich extracts were run on SDS-PAGE to visualize the approximate size and protein contents in these column fractions. Distinct bands of proteins were found in the water fractions for all three top hits, P00303, P00397, and R00137 (Figure 4). It was noted that there was interference in some of the high methanol fractions, likely because of water-insoluble material loaded into the wells of the gels causing streaking.

### 2.5. Activity of C18 Column-Fractionated AE Extracts

Aqueous extracts of the 11 top hits were fractionated on a C18 column with a methanol gradient before activity evaluation (Table 2).

All c18 column fractions were tested for PBMC cytotoxicity at 25 μg/mL. A measurable activity was observed for each of the fractionates (Figure 5). It was found that fractionated P00303-50ME had the highest activity for P00303 AE, while the P00303-100ME fractionate showed the least activity. In the case of P00397, the highest activity was located at P00397-WA-50ME and P00397-50ME. The lowest activity was observed at P00397-100ME. The P00137 methanol fractionate showed the highest activity for fractionate P00137-WA, while the lowest activity was observed at P00137-50-100ME.

### 2.6. Ultrafiltration of High-Protein Hits

The C18 column water fractions (WA) from the aqueous extracts of the top three hits were further investigated. The water fraction was ultrafiltered, as described in the methods, into three sub-fractions based on the molecular weights of <3 kDa, 3–30 kDa, and >30 kDa. These sizes were selected based on the majority of the proteins detected on the SDS-PAGE of the top hits. The yields from the ultrafiltration are shown in Table 3.

The ultrafiltered fractions were run on SDS-PAGE to ensure adequate protein segregation (Figure 6). The 3–30 kDa protein was found to be the prominent protein in all the top hits that were similar to the molecular weight distribution of Colostrum C18 column water fraction. The colostrum-based TF also contained proteins with molecular weights of 3–30 kDa and >30 kDa.

### 2.7. Activity of Ultrafiltrates

The ultrafiltrates were tested for their activity in the PBMC cytotoxicity assay at 12.5 μg/mL (Figure 7). This concentration was chosen in order to discriminate between the fractions with activity at higher concentrations that could be easily saturated in the PBMC assay. The AE of each top hit was tested at the same concentration as that of the ultrafiltrate. It was found that, in the case of P00303, the 3–30 kDa fraction was the one with the highest activity. This activity was better than the original AE at the same concentration and was also comparable to the positive controls TF and IL-2. All three fractions of P00397 showed comparable activities compared to the original AE which were comparable to the controls TF and IL-2. None of the fractions of P00137 showed activities better than the original AE or the controls C-TF and IL-2.

### 2.8. PBMC Cytotoxic Activity Confirmation

P00303 and P00397 were selected as the final leads due to their high protein content and high activity in the 3–30 kDa ultrafiltration fractions. The activity of these top leads was confirmed in a PBMC cytotoxicity assay using an extraction method focused on protein from the plant parts (i.e., seed) compared to the original AEs and the 3–30 kDa ultrafiltration fractions. To confirm activity, we extracted proteins from raw seed materials and tested them in the PBMC cytotoxicity assay at eight concentrations (0.19, 0.39, 0.78, 1.56, 3.13, 6.25, 12, and 25 μg/mL) (Figure 8). While higher concentrations showed lower activities due to interference with the lysis buffer, the cytotoxicity was relatively consistent for doses ranging from 0.19 to 6.25 µg/mL.

As a result of the strong and similar activities observed at 0.78 μg/mL in both P00303- and P00397-treated PBMCs, this concentration was chosen to compare to the original AEs and the 3–30 kDa ultrafiltration fractions. The P00303 protein extract at 0.78 μg/mL was compared to the original P00303 AE at 3.13 μg/mL and the P00303 AE-WA 3–30 kDa ultrafiltrate fraction at 3.13 μg/mL. The 0.78 μg/mL of protein is comparable to the level of protein calculated by the Bradford assay in the aqueous extract at 3.13 μg/mL. The same concentrations were tested for P00397. Ranges of effectiveness from 74.2to 77.8% and 75.2 to 79.0% were observed for P00303 and P00397, respectively. All the fractions, extracts, and purified proteins tested exhibited comparable or higher activities than C-TF, confirming their effectiveness in this assay (Figure 9).

### 2.9. Protein Identification

Proteins that were non-native to the plant that was queried (P00303, *R. sativus* or P00397, *B. juncea*) were assessed for homology to the queried species. This homology is listed on the right-most column of Table 4 and Table 5.

The 3–30 kDa fraction of P00303, Raphanus sativus, contained 17 proteins. Seven of these are known storage proteins, with five of them identified from *R. sativus* sequences and two from related species of the same genus plant with high homology to *R. sativus* (Table 4). There was one *R. sativus* defensin protein that accounted for 3% of the total protein in the sample, and there was a Kunitz trypsin inhibitor, which is a protease that protects the seed proteins from being degraded by their environment. The remaining nine proteins were low in abundance, and they included one cell wall adaptation protein and one cell signaling protein. Many of the others were hypothetical or uncharacterized, or they had known domains but unknown functions (the complete list of proteins is available in the Appendix A).

The 3–30 kDa fraction of P00397, *Brassica juncea*, had 61 proteins (Table 5). The seven most abundant proteins were all seed storage proteins, Napin, and 2S seed storage proteins. Allergen Bra j 1-E, the only protein found that is native to *B. juncea*, is also a 2S seed storage protein. There were a number of plant defense and stress response genes that were expressed, including Chitin-binding allergen Bra r 2, which is involved in PAMP-triggered immunity (PTI) in plants. The others were involved in defense against microorganisms or environmental stress (the complete list of proteins is available at the Appendix A).

## 3. Discussion

In the search for plant-based peptides that possess characteristics with the potential to increase immune responses, we screened a total of 100 protein-rich aqueous plant extracts in a PBMC cytotoxicity assay known to contain high percentages of NK cells. An aqueous extract was chosen understanding that the plant peptides are more likely to reside in the water extract than the organic extract. A colostrum-based transfer factor was used for comparative analysis. *Raphanus sativus* (radish) or *Brassica juncea* (mustard) were determined to have higher contents of protein and showed higher cytotoxic activities than the reference colostrum controls. Subsequent protein identification of ultrafiltrates from water extract with molecular mass 3–30 kDa, the top two leads, *Raphanus sativus* and *Brassica juncea,* were found to contain more storage protein than proteins involved in plant defense. There is a possibility that the enhanced NK cell cytotoxicity activity observed in the current screening could be partially explained by the presence of a family of peptides in radish and mustard with a defensive role.

Living organisms, ranging from microorganisms to vertebrates, invertebrates and plants, have evolved mechanisms to actively defend themselves against pathogens in their ecosystem. Vertebrates use a highly developed adaptive immune system to deploy active antibodies and trained killer cells to recognize and eliminate specific attackers [1]. In contrast, plants protect themselves through their innate immunity, a widespread defense strategy involving the production of antimicrobial peptides. While mammals, like humans, transfer antibodies through their breast milk to protect their offspring from disease early in life, plants transfer their protection against the pathogens and parasites to their seedlings through their DNA, which can provide lifelong protection not only to immediate offspring but also to subsequent generations [22]. It has been reported that the seeds of plants attacked by a pathogen were found to contain higher concentrations of chemical defense compounds that could provide augmented protection to the offspring. For example, when a tobacco plant, *Nicotiana tabacum*, was exposed to tobacco mosaic virus, the plant’s progeny was found to be more resistant not only to the virus but also to some bacteria and molds [23].

Defense-related proteins, i.e., defensin-like protein 192, Kunitz trypsin inhibitor 1, chitin-binding allergen Bra r 2, and pathogenesis-related thaumatin superfamily protein, have been identified in the top two leads (radish and mustard) of the current screening. Plant defensins are families of peptides that make up part of the innate immune system of plants directed against phytopathogens [24]. They are structurally and functionally related to defensins that have been previously characterized in mammals and insects with molecular masses ranging between 5 kDa and 7 kDa [25].

Most plant defensins are seed-derived [26]. For instance, it has been reported that, in radish, defensin proteins (Raphanus sativus-antifungal protein, Rs-AFPs, 5-kD cysteine-rich proteins) represent 0.5% of the total protein in seeds and were functionally found to provide a favorable microenvironment to the seedling at the time of germination by suppressing soil fungal growth [27]. A similar anti-fungal peptide (AFP1) that shares 100% amino acid sequence identity with *Raphanus sativus* defensin (Rs-AFP1) has also been reported from the seed of *B. juncea* [28].

Plant defensins have been found to have immunologic activities in inhibiting infection and inducing apoptosis in human pathogens. For example, induction of apoptosis and activation of caspases or caspase-like proteases in the human pathogen *Candida albicans* have been observed in radish antifungal plant defensin RsAFP2 [29]. Similarly, anti-breast cancer and leukemia cells’ proliferative activity and immunodeficiency-virus-type 1 reverse transcriptase inhibitory activities in ground bean antimicrobial peptides (AMPs) have also been reported [30]. 

The cross kingdom defensin biological effect was also investigated in transgenic plants that express the human defensin. Since human beta-defensins and plant defensins share structural homology, functional homology between these defensins of different kingdoms was tested. *Arabidopsis thaliana* plants expressing human beta-defensin-2 (hBD-2) were found to be more resistant against the broad-spectrum fungal pathogen *Botrytis cinerea* and that the resistance was correlated with the level of active hBD-2 produced in these transgenic plants [31].

Higher proportions of storage proteins than defense proteins were found in the top two leads, *B. juncea* and *R. sativus*. Seed storage proteins accounted for the bands at 12, 14, and 19–20 kDa, and the next group containing Kunitz trypsin inhibitor 1, uncharacterized protein Rs2_04757, and the hypothetical proteins likely accounted for the band at 22–24 kDa in SDS-PAGE. Those were the four major bands in the fraction for radish (P00303). It is likely that although cruciferin was predicted to be present in the seeds, it was denatured and precipitated during one of the extraction steps, as it is not very thermally stable.

*B. juncea* is not as well characterized as *R. sativus*. The Brassicaceae taxonomy only contains 3183 *B. juncea* genes, whereas it contains 115,890 *R. sativus* genes and 38,740 *H. incana* genes. So many of the genes that came up from the *B. juncea* query were from better-characterized species of Brassicaceae. Many of the genes did not have putative homologs in *B. juncea*, suggesting the possibility that these genes could be not yet characterized in *B. juncea*. Many of the low-abundance proteins were uncharacterized, hypothetical proteins or domain-containing proteins of unknown or various functions. Many of the others were involved in such processes as vesicle trafficking and endocytosis, cytoskeleton remodeling, and protein degradation. It is unlikely that proteins with molecular weights of much greater than 30 kDa are actually present in the sample and may result from a low-fidelity peptide sequence, since this sample was ultrafiltered to exclude proteins greater than 30 kDa.

Collectively, the data depicted here suggest that water extracts of *Raphanus sativus* or *Brassica juncea* could be considered for further characterization and immune functional exploration with a possibility of dietary supplemental use to bolster human recipients’ immune response. If clinically proven, the current report could expand the application of plant-based peptides as an additional safe and efficacious source of oral supplements with immune transfer activity in humans.

## 4. Materials and Methods

### 4.1. Plant Selection

A total of 100 diverse plant aqueous extracts were selected based on their protein contents following a literature review. Plant parts such as root, stem, bark, fruit, seed, leaf, gum resin, aerial part, trunk bark, nut, leaf-stem, seed casing, cob, corn silk and whole plants were used for screening.

### 4.2. Preparation of Aqueous Extracts

Dried ground plant powder of each plant (20 g) was loaded into 100 mL stainless steel tubes and extracted twice with an organic solvent mixture (methylene chloride/methanol in a ratio of 1:1) using an ASE 300 automatic extractor at 80 °C and 1500 psi pressure. We carried out this step as our standard procedure to prepare extracts for our extract library (including organic extracts and aqueous extracts). The extract solution was automatically filtered and collected, then the plant powder was flushed with fresh solvent and purged with nitrogen gas to dry before switching to aqueous extraction (DI water) at 50 °C. The aqueous solution was filtered and freeze-dried to provide an aqueous extract (AE). We directly used the aqueous extracts from the library for this study.

### 4.3. C18 Fractionation

Plant aqueous extract (AE) was dissolved in 13 mL of DI water and 2 mL of DMSO (to improve the solubility). The solution was loaded onto a pre-packed Biotage^®^ Sfär C18 column (Biotage San Jose, CA, USA) (Duo 100 Å 30 µm, 60 g) and pushed into the column bed. Liquid dripping from the column was collected into a waste beaker until solution just reached the frit. The column then was eluted with a gradient of methanol in water as follows: 100% DI water, 2 column volume (CV); 100% DI water to 50% methanol, 1.4 CV; 50% methanol, 1.3 CV; 50% methanol to 100% methanol, 1.6 CV and 100% methanol, 2 CV. The elute was collected into test tubes and combined into 5 fractions: WA, WA-50ME, 50ME, 50–100ME, and 100ME. Solvents were evaporated with a rotary vacuum evaporator and the fractions were dried to yield column fractions.

### 4.4. Ultrafiltration

The selected reverse phase column fractions were ultrafiltered as follows. A total of 250 mg column fraction dried sample was dissolved in 125 mL of DI water. Each membrane disc was rinsed by floating its skin (glossy) side down in a beaker with DI water and sonicating for at least 1 h, changing the water 3 times. The 30 kDa membrane disc was placed in an ultrafiltration device with the skin (glossy) side up toward the solution. The solution from step 1 was transferred into the ultrafiltration device and nitrogen gas was applied to begin collecting the <30 kDa fraction that passed through the membrane disk. Once completed, the membrane was removed and placed in a beaker with ~250 mL of DI water. It was sonicated for ~1 h and the fraction was kept as the >30 kDa fraction. The membrane in the ultrafiltration device was changed to 3 kDa and the <30 kDa fraction was transferred to filter it through the new membrane. Once completed, the membrane was removed and placed in a beaker with ~250 mL of DI water. It was sonicated for ~1 h and the fraction was kept as the 3–30 kDa fraction. The filtrate solution that passed through the 3 kDa membrane was labeled as the <3 kDa fraction. All the ultrafiltration fractions were freeze-dried to remove water and to obtain powdered materials.

### 4.5. Bradford Assay

A volume of 5 μL of standards and plant AE samples was added to 250 μL of Bradford reagent (Bio-Rad, 5000202) (Bio-Rad Laboratories, Hercules, CA, USA). The assay was incubated for five minutes at room temperature before the absorbance was read at 595 nm. The protein concentration of the aqueous extracts was calculated using a bovine serum albumin standard curve.

### 4.6. SDS-PAGE

The sample was denatured in SDS sample buffer (100 mM Tris, pH 6.8, 20% glycerol, 4% SDS 0.05% bromophenol blue, and 10% β-mercaptoethanol) and boiled for 5 min. A 5–15% Tris-Tricine gradient gel was used for resolving the proteins. The gel was stained with Coomassie (45% methanol, 10% glacial acetic acid, 0.25% Coomassie G-250) for 1 h nutating at room temperature, and it was destained three times for 30 min each with 45% methanol and 10% glacial acetic acid. The gel was visualized on ThermoFisher iBright gel and a Western blot documentation station.

### 4.7. Protein Identification

LC-MS/MS analysis and protein sequencing were carried out on 3–30 kDa ultrafiltration fractions from P00303 and P00397 at Applied Biomics (Hayward, CA, USA) for protein identification. Briefly, the proteins were reduced and alkylated, trypsin digested, and subjected to nano LC-MS/MS. Homologous protein sequences were searched using the Brassicaceae taxonomy database from the National Center for Biotechnology Information at the National Institutes of Health and a list of the proteins identified was generated.

emPAI was calculated according to previously reported methods [32]. emPAI% was calculated to be the percent abundance of the specified protein in the sample. Molecular weight information and percent homology were obtained from the National Center for Biotechnology Information at the National Institutes of Health. Briefly, for P00303, the *R. sativus*, non-*R. sativus* proteins that were identified were checked for homology to *R. sativus* proteins using Protein BLAST. For P00397, the *B. juncea* and non-*B. juncea* proteins that were identified were checked for homology to *B. juncea* proteins.

### 4.8. PBMC Cytotoxicity Assay

The PBMC cytotoxicity assay with image cytometry was adapted from the literature [33]. Briefly, human peripheral blood mononuclear cells (PBMCs, Millipore Sigma, 0002145) were plated in 96-well cell culture plates at 250,000 cells/well. For screening, these cells were treated with 50 and 100 μg/mL extracts overnight in parallel with untreated cells. Cells treated with a colostrum-based transfer factor at 500 μg/mL and IL-2 at 20 μg/mL were used as references. The next day, one million human cancerous K562 lymphoblasts were resuspended in 2.5 mL of cell culture media and 2.5 mL of 10 μM calcein-AM in a 15 mL conical flask, which was inverted to mix and incubated at 37 °C for 30 min. The K562 cells were centrifuged at 3000 rpm for five minutes to forms pellets and resuspended in fresh media. They were pelleted and resuspended three times to wash out unbound calcein-AM. K562 cells were added to PBMCs at a density of 10,000 cells/well (a 25:1 effector/target cell ratio). The 96-well plate was centrifuged at 600 rpm for two minutes to gently settle the suspended cells on the plate bottom. The plates were scanned for green fluorescent calcein-AM on an ImagExpress Pico by Molecular Devices every hour for 4 h. The parameters were adjusted to detect single bright green cells, and the number of live cells was calculated at each time point. The number of surviving K562 cells was normalized to the untreated control and the IL-2 control to reduce the variation among individual experiments.

## 5. Patents

A provisional patent has been filed with title Plant-Based Composition That Modulates Cellular Immunity by inventor Dr. David Vollmer.

## Figures and Tables

**Figure 1 molecules-28-07961-f001:**
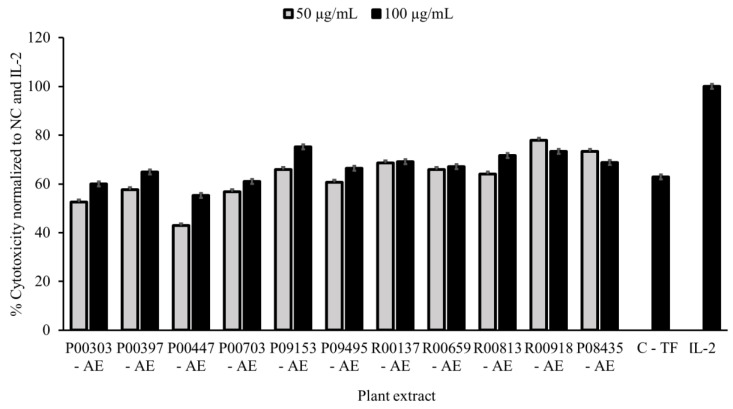
Cytotoxicity data as a measure of activity for primary screening. PBMC cells (250,000 cells/well) in a 96-well plate were treated with 50 and 100 μg/mL aqueous plant extracts overnight in parallel with untreated cells. Cells treated with colostrum-based transfer factor (C-TF) at 500 μg/mL, and IL-2 at 20 μg/mL were used as references. K562 cells were then added to PBMCs at a density of 10,000 cells/well at 25:1 effector/target cell ratio. The plates were scanned for green, fluorescent calcein-AM on an ImagExpress Pico every hour for 4 h. The number of surviving K562 cells was normalized to the untreated control and the IL-2 control. Data are expressed as percent cytotoxicity. NC = normal control.

**Figure 2 molecules-28-07961-f002:**
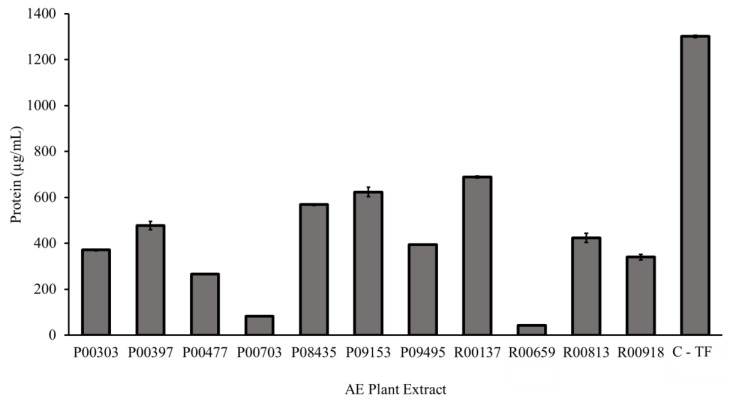
Protein content of 11 primary hits from aqueous plant extracts as measured by a Bradford assay. A Bradford assay was used to determine the concentration of proteins in each AE plant extract compared to C-TF. Both 5 μL of the standard and 5 μL of plant AE sample were added to 250 μL Bradford reagent. The assay was incubated for five minutes at room temperature before the absorbance was read at 595 nm. The protein concentration of the aqueous extracts was calculated using a bovine serum albumin standard curve.

**Figure 3 molecules-28-07961-f003:**
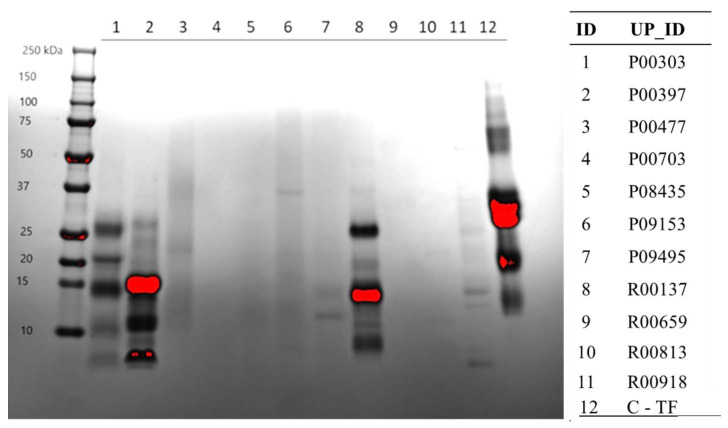
Confirmation of protein content from the AE extracts of the 11 primary hits on SDS page. SDS-PAGE gel visualized the protein content of each plant extract denatured in SDS sample buffer and resolved by Tris-Tricine gradient gel. The gel was stained with Coomassie for 1 h and destained three times for 30 min each with 45% methanol and 10% glacial acetic acid. The gel shows that the extracts with the highest protein content in the range of the gel were P00303, P00397, R00137, and C-TF.

**Figure 4 molecules-28-07961-f004:**
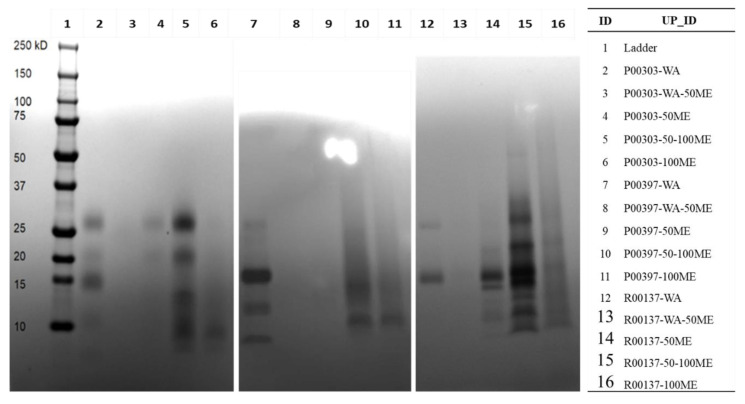
Protein content and size of top three hits from C18 column fractions. SDS-PAGE gel visualized the protein content of the indicated methanol fractions denatured in SDS sample buffer and resolved by Tris-Tricine gradient gel. The gel was stained with Coomassie for one hour and destained three times for 30 min each with 45% methanol and 10% glacial acetic acid. WA: water; WA-50ME: water to 50% MeOH; 50ME: 50% MeOH; 50–100ME: 50–100% MeOH; 100ME: 100% MeOH.

**Figure 5 molecules-28-07961-f005:**
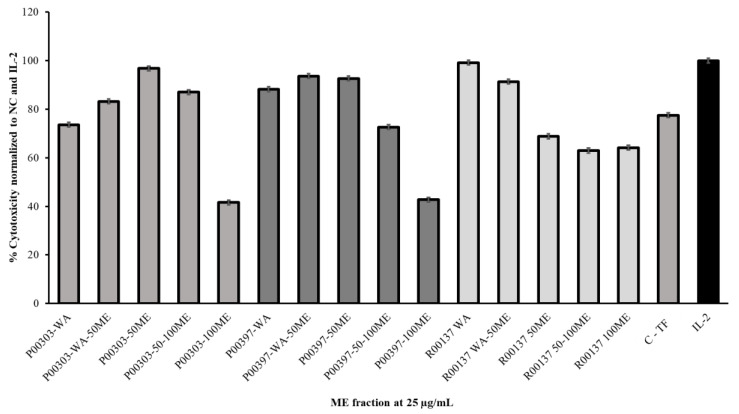
Cytotoxic effects of top hit C18-column-fractionated AEs on calcein-AM-stained target cells normalized to the negative and IL-2 controls. Samples were tested at 25 μg/mL. C-TF (500 µg/mL) and IL-2 (20 µg/mL). A PBMC cytotoxicity assay using human cancerous lymphoblast cells (K562 cells) as a target was utilized. WA: water; WA-50ME: water to 50% MeOH; 50ME: 50% MeOH; 50–100ME: 50%-100% MeOH; 100ME: 100% MeOH.

**Figure 6 molecules-28-07961-f006:**
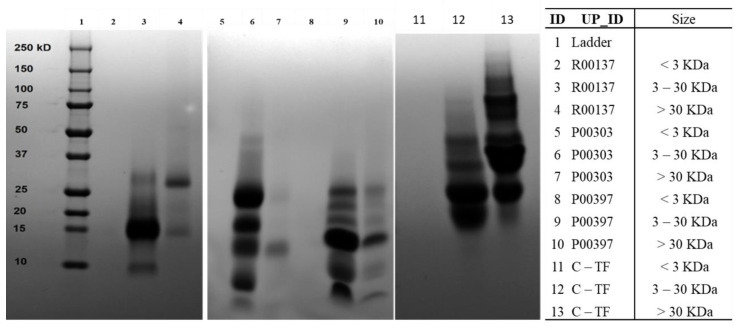
SDS-PAGE gel protein content of ultrafiltrate fractions from the WA fraction of top three AE hits (R00137, P00303, and P00397). A 250 mg dried sample from a column fraction was dissolved in 125 mL DI water and used for ultrafiltration. Membrane discs of 3 kDa and 30 kDa were used for size segregation of ultrafiltrates. All the ultrafiltered fractions were freeze-dried to remove water and get powdered materials. Each hit was tested at <3 kDa, 3–30 kDa and > 30kDa.

**Figure 7 molecules-28-07961-f007:**
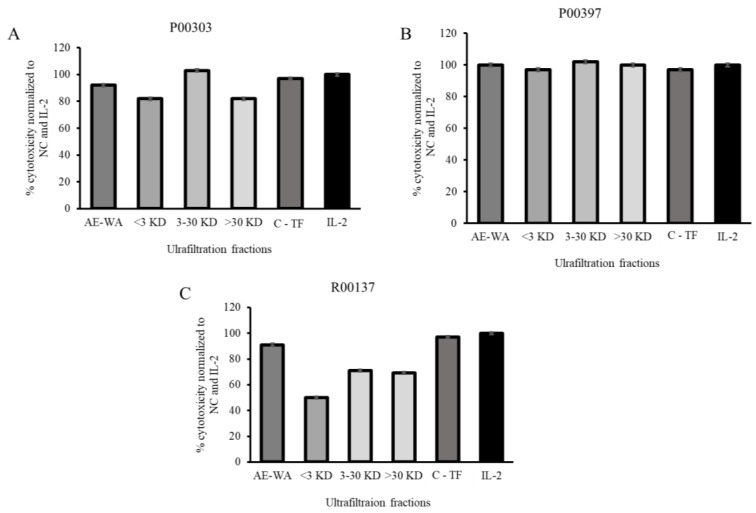
Cytotoxic effects of the three ultrafiltration fractionated samples on calcein-AM-stained target cells normalized to the negative C-TF (500 µg/mL) and IL-2 (20 µg/mL) controls. The ultrafiltrates were tested for their activity in the PBMC cytotoxicity assay at 12.5 μg/mL. (**A**) P00303; (**B**) P00397, (**C**) R00137.

**Figure 8 molecules-28-07961-f008:**
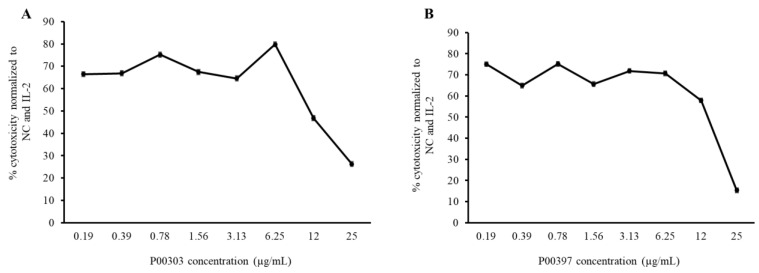
Cytotoxicity dose curve of protein from P00303 (**A**) and P00397 (**B**) seed extracts. A PBMC cytotoxicity assay using human cancerous lymphoblast cells (K562 cells) as a target was carried out at eight concentrations (0.19, 0.39, 0.78, 1.56, 3.13, 6.25, 12, and 25 μg/mL).

**Figure 9 molecules-28-07961-f009:**
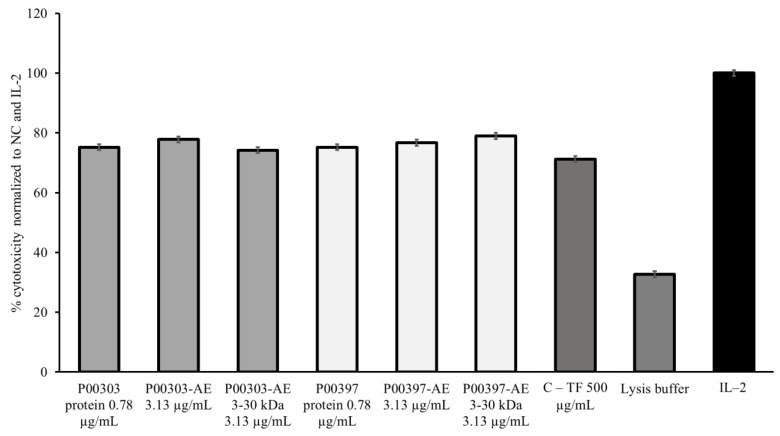
Cytotoxic effects of the ultrafiltrate fractionated samples from P00303 and P00397 compared to protein extracts on calcein-AM-stained target cells normalized to the negative C-TF (500 µg/mL) and IL-2 (20 μg/mL) controls. The P00303 and P00397 protein extracts at 0.78 μg/mL were compared to the original P00303/ P00397 AEs at 3.13 μg/mL and the P00303/P00397 AE-WA 3–30 kDa ultrafiltrate fraction at 3.13 μg/mL. A PBMC cytotoxicity assay using human cancerous lymphoblast cells (K562 cells) as a target was used for the comparison.

**Table 1 molecules-28-07961-t001:** Primary screening hits of aqueous extracts tested in a PBMC assay.

NO.	UP_ID	Species	Common Names	Part	Extract *
1	P00303	*Raphanus sativus*	Radish	Seed	AE
2	P00397	*Brassica juncea*	Mustard greens	Seed	AE
3	P00477	*Dendrocalamus strictus*	Calcutta Bamboo	Seed	AE
4	P00703	*Alstonia scholaris*	Blackboard tree	Bark	AE
5	P08435	*Sorghum bicolor*	Great millet	Leaf-Stem	AE
6	P09153	*Solanum incanum*	Bitter apple	Fruit	AE
7	P09495	*Zea mays*	Corn	Corn silk	AE
8	R00137	*Cocos nucifera*	Coconut	Fruit meat	AE
9	R00659	*Beta vulgaris*	Beet	Root	AE
10	R00813	*Taraxacum officinale*	Common dandelion	Leaf	AE
11	R00918	*Psoralea corylifolia*	Babchi	Fruit	AE

* AE—aqueous extracts were tested at 50 and 100 μg/mL, in duplicate, in a PBMC assay incubated overnight with water as a vehicle, and IL-2 (20 μg/mL) and colostrum-based TF (C-TF hereafter) (500 μg/mL) as controls.

**Table 2 molecules-28-07961-t002:** Yields from the C18 column fractionation of the three selected top hits.

ID	Species	Part	Extract (mg)	WA (mg)	WA-50ME	50ME (mg)	50–100ME	100ME (mg)
C-TF	-	-	1017.3	1342.0	57.7	21.2	7.4	8.2
P00303	*Raphanus sativus*	seed	2657.0	2889.5	134.5	285.7	40.7	17.7
P00397	*Brassica juncea*	seed	2083.7	2959.4	120.5	133.4	16.6	14.6
R00137	*Cocos nucifera*	endosperm	894.4	468.7	54.6	29.6	6.5	7.6

WA: water; WA-50ME: water to 50% MeOH; 50ME: 50% MeOH; 50–100ME: 50–100% MeOH; 100ME: 100%.

**Table 3 molecules-28-07961-t003:** Yields from selective molecular weight ultrafiltration of C18 column water fractions.

WA Fraction ID	<3 kDa (mg) (%)	3–30 kDa (mg) (%)	>30 kDa (mg) (%)
P00303-AE-WA	146.5 (93%)	4.0 (2.5%)	5.7 (3.6%)
P00397-AE-WA	145.0 (86%)	22.0 (13%)	2.3 (1.4%)
R00137-AE-WA	191.3 (95%)	5.4 (2.7%)	4.5 (2.2%)
C-TF-WA	151.1 (85%)	6.0 (3.4%)	19.9 (11%)

**Table 4 molecules-28-07961-t004:** Proteins in P00303 *Raphanus sativus* 3–30 kDa fraction sorted from most to least abundant.

Accession No.	Protein Name	emPAI	emPAI %	MW (kDa)	% Homology to *R. sativus*	Function
gi|75107016	Napin-3 (*Brassica napus*)	2.31	20.68%	14	76%	Seed storage
gi|2440726951	2S seed storage protein 4 (*Raphanus sativus*)	2.11	18.89%	20	…	Seed storage
gi|2440684703	2S seed storage protein 4 (*Raphanus sativus*)	2.09	18.71%	20	…	Seed storage
gi|2440727854	2S seed storage protein 4 (*Raphanus sativus*)	0.77	6.89%	20	…	Seed storage
gi|2440715477	2S seed storage protein 4 (*Raphanus sativus*)	0.76	6.80%	20	…	Seed storage
gi|2334152968	Bifunctional inhibitor/plant lipid transfer protein/seed storage helical domain-containing protein (*Hirschfeldia incana*)	0.61	5.46%	12	96%	Seed storage
gi|2440706739	Defensin-like protein 192 (*Raphanus sativus*)	0.34	3.04%	19	…	Plant defense
gi|2440694918	2S seed storage protein 4 (*Raphanus sativus*)	0.33	2.95%	20	…	Seed storage
gi|2440692911	kunitz trypsin inhibitor 1 (*Raphanus sativus*)	0.30	2.69%	22	…	Plant defense
gi|2440724679	Uncharacterized protein Rs2_04757 (*Raphanus sativus*)	0.29	2.60%	22	…	Unknown
gi|2440690659	hypothetical protein Rs2_41200 (*Raphanus sativus*)	0.27	2.42%	24	…	Unknown
gi|2440695549	hypothetical protein Rs2_38119 (*Raphanus sativus*)	0.26	2.33%	24	…	Unknown
gi|2334151640	Expansin (*Hirschfeldia incana*)	0.23	2.06%	28	28%	Cell wall adaptation
gi|2440696312	Receptor-like serine/threonine-protein kinase SD1-8 (*Raphanus sativus*)	0.15	1.34%	41	…	Cell signaling
gi|2440721862	Putative F-box protein (*Raphanus sativus*)	0.14	1.25%	43	…	Various
gi|2440686810	hypothetical protein Rs2_45685 (*Raphanus sativus*)	0.11	0.98%	56	…	Unknown
gi|2440688317	Octicosapeptide/Phox/Bem1p (PB1) domain-containing protein/TPR-containing protein (*Raphanus sativus*)	0.10	0.90%	65		Various

Proteins were reduced and alkylated, trypsin digested, and subjected to nano LC-MS/MS. Homologous protein sequences were searched using the Brassicaceae taxonomy database from the National Center for Biotechnology Information at the National Institutes of Health and a list of the proteins identified was generated. Full characterizations of peptides have been provided in the Appendix A.

**Table 5 molecules-28-07961-t005:** Proteins in the P00397 Brassica juncea 3–30 kDa fraction sorted from most to least abundant.

Accession No.	Protein Name	emPAI	emPAI %	MW (kDa)	% Homology to *B. juncea*	Function
gi|75107016	Napin-3 (*Brassica napus*)	6.90	19.76%	14.04	79%	storage
gi|32363444	Allergen Bra j 1-E (*Brassica juncea*)	6.21	17.78%	14.65	…	storage
gi|2440684703	2S seed storage protein 4 (*Raphanus sativus*)	4.44	12.71%	20.14	84%	storage
gi|2440727853	2S seed storage protein 4 (*Raphanus sativus*)	2.23	6.39%	20.83	85%	storage
gi|112747	Napin embryo-specific (Brassica napus)	2.19	6.27%	21.02	82%	storage
gi|2440727854	2S seed storage protein 4 (*Raphanus sativus*)	1.66	4.75%	19.94	85%	storage
gi|2440715477	2S seed storage protein 4 (*Raphanus sativus*)	1.65	4.73%	19.97	82%	storage
gi|32363456	Chitin-binding allergen Bra r 2 (*Brassica rapa*)	1.59	4.55%	10.05	54%	Plant defense
gi|2334156319	Em-like protein GEA6 (*Hirschfeldia incana*)	0.64	1.83%	9.62	98%	Seed desiccation
gi|2440698549	Protein kinase superfamily protein (*Raphanus sativus*)	0.49	1.40%	12.02	None	Various
gi|2334145742	Uncharacterized protein HA466_0114050 (*Hirschfeldia incana*)	0.46	1.32%	12.65	None	Unknown
gi|2334145261	Ubiquitin-like domain-containing protein (*Hirschfeldia incana*)	0.39	1.12%	14.68	99%	Protein degradation
gi|2440705679	hypothetical protein Rs2_30933 (*Raphanus sativus*)	0.33	0.95%	16.65	None	Unknown
gi|2334150478	Protein C2-DOMAIN ABA-RELATED 6 (*Hirschfeldia incana*)	0.31	0.89%	18.16	None	Stress response
gi|2440696237	Uncharacterized protein Rs2_38807 (*Raphanus sativus*)	0.30	0.86%	18.46	None	Unknown
gi|2334153535	Superoxide dismutase (*Hirschfeldia incana*)	0.25	0.72%	21.67	64%	Antioxidation
gi|2440703857	Protein kinase domain protein (*Raphanus sativus*)	0.25	0.72%	21.60	42%	Various
gi|2334159199	hypothetical protein HA466_0029400 (*Hirschfeldia incana*)	0.22	0.63%	24.38	44%	Unknown
gi|2440688376	hypothetical protein Rs2_44277 (*Raphanus sativus*)	0.21	0.60%	25.27	None	Unknown
gi|2440705714	Pathogenesis-related thaumatin superfamily protein (*Raphanus sativus*)	0.20	0.57%	27.09	34%	Plant defense
gi|2440692471	PHD finger protein ALFIN-LIKE 6 (*Raphanus sativus*)	0.19	0.54%	27.77	None	Stress response
gi|2334151640	Expansin (*Hirschfeldia incana*)	0.18	0.52%	27.91	30%	Cell wall adaptation
gi|2440685386	Myrosinase 2 (*Raphanus sativus*)	0.17	0.49%	30.84	93%	Plant defense
gi|2440689373	Helitron-like N domain-containing protein (*Raphanus sativus*)	0.16	0.46%	32.31	94%	Gene transposase
gi|2440719378	Mitochondrial uncoupling protein 1 (*Raphanus sativus*)	0.16	0.46%	32.47	None	Antioxidation
gi|2334151314	E3 ubiquitin-protein ligase RHC1A (*Hirschfeldia incana*)	0.15	0.43%	35.83	None	Protein degradation
gi|2440695821	hypothetical protein Rs2_38391 (*Raphanus sativus*)	0.14	0.40%	38.70	None	Unknown
gi|2440678052	FAD-dependent oxidoreductase family protein (*Raphanus sativus*)	0.13	0.37%	39.62	None	Amino acid biology
gi|2440680072	Tubby-like F-box protein 10 (*Raphanus sativus*)	0.13	0.37%	41.35	33%	Stress response
gi|2440692001	Lipase class 3-related protein (*Raphanus sativus*)	0.13	0.37%	40.28	None	Lipid metabolism
gi|2440700075	Protein kinase superfamily protein (*Raphanus sativus*)	0.13	0.37%	40.87	None	Various
gi|2440705359	DNAJ heat shock N-terminal domain-containing protein (*Raphanus sativus*)	0.13	0.37%	41.18	52%	Stress response
gi|2440681029	FAD/NAD(P)-binding oxidoreductase family protein (*Raphanus sativus*)	0.12	0.34%	44.88	None	Amino acid biology
gi|2440697078	hypothetical protein Rs2_32681 (*Raphanus sativus*)	0.12	0.34%	43.46	None	Unknown
gi|2440697973	Protein RESTRICTED TEV MOVEMENT 2 (*Raphanus sativus*)	0.12	0.34%	43.38	None	Stress response
gi|2440691863	Mitochondrial transcription termination factor family protein (*Raphanus sativus*)	0.11	0.32%	47.22	None	Gene regulation
gi|2440701438	hypothetical protein Rs2_26692 (*Raphanus sativus*)	0.11	0.32%	49.30	None	Unknown
gi|2440680808	RNA-binding (RRM/RBD/RNP motifs) family protein (*Raphanus sativus*)	0.10	0.29%	51.49	27%	Transcription/ translation
gi|2440716970	Pectin lyase-like superfamily protein (*Raphanus sativus*)	0.10	0.29%	50.06	None	Cell wall adaptation
gi|2440719117	IQ-domain 3 (*Raphanus sativus*)	0.10	0.29%	49.60	None	Calcium signaling
gi|2440724245	putative F-box protein (*Raphanus sativus*)	0.10	0.29%	52.44	36%	Various
gi|2440686810	hypothetical protein Rs2_45685 (*Raphanus sativus*)	0.09	0.26%	56.45	None	Unknown
gi|2334150521	Uncharacterized protein HA466_0085320 (*Hirschfeldia incana*)	0.09	0.26%	55.38	44%	Unknown
gi|2440685993	Ypt/Rab-GAP domain of gyp1p superfamily protein (*Raphanus sativus*)	0.09	0.26%	58.83	None	Vesicle trafficking
gi|2440688005	Aldehyde dehydrogenase family 2 member C4 (*Raphanus sativus*)	0.09	0.26%	54.28	None	Stress response
gi|2440731844	Uncharacterized protein Rs2_02820 (*Raphanus sativus*)	0.08	0.23%	61.32	None	Unknown
gi|2440693227	Sulfate transporter 1.2 (*Raphanus sativus*)	0.07	0.20%	71.79	96%	Mineral uptake
gi|2440700872	hypothetical protein Rs2_26126 (*Raphanus sativus*)	0.07	0.20%	74.34	None	Unknown
gi|2440678058	DNA polymerase alpha catalytic subunit (*Raphanus sativus*)	0.06	0.17%	87.06	None	DNA replication
gi|2440685990	Phosphatidylinositol 4-phosphate 5-kinase 6 (*Raphanus sativus*)	0.06	0.17%	81.50	None	Endocytosis
gi|2440706207	hypothetical protein Rs2_31461 (*Raphanus sativus*)	0.06	0.17%	84.60	None	Unknown
gi|2440732392	tRNAse Z3 (*Raphanus sativus*)	0.06	0.17%	89.65	None	tRNA synthesis
gi|2334151474	ENTH domain-containing protein (*Hirschfeldia incana*)	0.05	0.14%	98.92	None	Vesicle transport
gi|2334153854	Protein NETWORKED 2D (*Hirschfeldia incana*)	0.05	0.14%	107.60	36%	Cytoskeleton
gi|2334157830	Leucine-rich repeat receptor-like serine/threonine-protein kinase BAM3 (*Hirschfeldia incana*)	0.05	0.14%	109.52	None	Root and leaf growth
gi|2440683401	Lon protease-like protein 1 (*Raphanus sativus*)	0.05	0.14%	109.31	None	Protein degradation
gi|2440684207	Villin-5 (*Raphanus sativus*)	0.05	0.14%	94.83	None	Cytoskeleton
gi|2440702903	hypothetical protein Rs2_28157 (*Raphanus sativus*)	0.05	0.14%	100.50	79%	Unknown
gi|2334154967	PHD-type domain-containing protein (*Hirschfeldia incana*)	0.04	0.11%	136.74	38%	Various
gi|2440711253	Leucine-rich repeat transmembrane protein kinase (*Raphanus sativus*)	0.04	0.11%	114.72	None	Stress response
gi|2440696774	E3 ubiquitin-protein ligase PRT6 (*Raphanus sativus*)	0.02	0.06%	221.34	None	Protein degradation

Proteins were reduced and alkylated, trypsin digested, and subjected to nano LC-MS/MS. Homologous protein sequences were searched using the Brassicaceae taxonomy database from the National Center for Biotechnology Information at the National Institutes of Health and generated a list of the proteins identified. Full characterizations of peptides have been provided in the Appendix A.

## Data Availability

Conclusions were made based on data depicted on this manuscript and Appendix A.

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
