# Peer review of "Discovery of Transfer Factors in Plant-Derived Proteins and an In Vitro Assessment of Their Immunological Activities"

_molecules, 2023, doi:10.3390/molecules28247961_

Round 1

Reviewer 1 Report

Comments and Suggestions for Authors

Make sure that the scientific names are mentioned in italics. Other specific comments can be viewed in the attached files

Comments on the Quality of English Language

Language is fine

Author Response

Thank you for the constructive feedback. Please refer the attached for our responses.  

Reviewer 2 Report

Comments and Suggestions for Authors

·        The assessment of transfer factors in proteins derived from plants is presented in an intriguing paper. The authors specifically looked for plant-based proteins or peptides that function as transfer factors to increase immunity. Despite being a well-written text, I discovered a few grammar mistakes all throughout. The English needs to be fixed moderately.

·        Names of genera and species need to be italicized.

·        I am worried about the AEs used in this study. High abundance/real proteins with high transfer factors to promote immunity may have been lost during the preparation of the protein lysate due to their weak water solubility. If there are any other solubilization techniques (such as Tris, PBS, etc.) that can cover a larger range of proteins that would be viable choices, that would be interesting to know about. 

·        The manuscript lacks specific statistics. To offer readers more confidence in the manuscript, I advise the authors to include "p value" and "n" in it.

·        Cite references for Protein Identification (i.e. emPAI analysis).

·        The results section presents the LC-MS/MS methodology. It would be helpful if the authors could go into more detail about it in the materials and methods section (e.g., instrument model and manufacture, column specifics, fundamental protein identification parameters, etc.). Additionally, I generally believe that if use LC-MS/MS, the MW limit might not be required. Even proteins with low molecular weight can be found using LC-MS/MS. You were not required to do the MW cutoff to extract the proteins. You never know what low molecular proteins you are missing.

·        The preparation of the protein worries me a little. After ultra-filtration, I am unable to detect any high molecular weight proteins in the provided figures. I can see how additional processing could cause many or a few great candidates to degrade. Lowering the processing stages may increase the likelihood of maintaining protein integrity without causing protein degradation, in my opinion. In addition, I would like to know if the authors included protease inhibitors before preparing the sample. I cannot locate the protease inhibitor details in the manuscript. For instance, no proteins are present in lanes 4, 5, 9, and 10 in Figure 3 or in lanes 3, 8, and 13 in Figure 4. It's obvious that most of the proteins are being lost.

·        To demonstrate that the proteins were similarly loaded onto the SDS-PAGE, I advise authors to do a western blot utilizing common housekeeping proteins. I've read over the sample preparation procedure. Authors have started with equal input raw material. However, because you could lose the proteins, processing stages like filtering, MW cutoff, etc., occasionally make a difference. Once more, sample preparation is crucial. I anticipate earning a point for using this approach. Again, the sample preparation matters a lot. I am expecting a valid point for using this method.

·        Data from LC-MS/MS must be submitted to the proteome consortium.  

Comments on the Quality of English Language

It looks fine to me in a general sense. Moderate editing is required. 

Author Response

Date: November 17, 2023

Dear Editor,

We express our utmost appreciation and gratitude to the editorial office and reviewers for their time and invaluable constructive comments. We have revised the full manuscript according to the suggestions and incorporated our response at appropriate sections of the body as follows:

Reviewer 2

  1. The assessment of transfer factors in proteins derived from plants is presented in an intriguing paper. The authors specifically looked for plant-based proteins or peptides that function as transfer factors to increase immunity. Despite being a well-written text, I discovered a few grammar mistakes all throughout. The English needs to be fixed moderately.

Reply: Manuscript underwent English language editing before submission. We have revised the manuscript as suggested for grammar.

  1. Names of genera and species need to be italicized.

Reply: Manuscript has been revised as suggest for italics.

  1. I am worried about the AEs used in this study. High abundance/real proteins with high transfer factors to promote immunity may have been lost during the preparation of the protein lysate due to their weak water solubility. If there are any other solubilization techniques (such as Tris, PBS, etc.) that can cover a larger range of proteins that would be viable choices, that would be interesting to know about. 

Reply: We agree with the reviewer’s comment. Weak water-soluble proteins could have been missed in the current methodology. We will incorporate the suggested solubilization techniques in our future work. Since our sample handling was uniform for all the materials, we pursued those with significant amount of proteins as hits.

  1. The manuscript lacks specific statistics. To offer readers more confidence in the manuscript, I advise the authors to include "p value" and "n" in it.

Reply: For this project, percent changes were considered as a better representation for the changes in cytotoxicity relative to controls. Statistical analysis for p-values could not be determined as comparison between a single plant extract and control in efficacy was the relative difference in presence or absence of cytotoxicity.

  1. Cite references for Protein Identification (i.e. emPAI analysis).

Reply: we have sited Ishihama Y, Oda Y, Tabata T, Sato T, Nagasu T, Rappsilber J, Mann M. Exponentially modified protein abundance index (emPAI) for estimation of absolute protein amount in proteomics by the number of sequenced peptides per protein. Mol Cell Proteomics. 2005 Sep;4(9):1265-72.

  1. The results section presents the LC-MS/MS methodology. It would be helpful if the authors could go into more detail about it in the materials and methods section (e.g., instrument model and manufacture, column specifics, fundamental protein identification parameters, etc.). Additionally, I generally believe that if use LC-MS/MS, the MW limit might not be required. Even proteins with low molecular weight can be found using LC-MS/MS. You were not required to do the MW cutoff to extract the proteins. You never know what low molecular proteins you are missing.

Reply: We have moved the methodology to the material and method sections. The analysis was carried out by external CRO as indicated in the section. The methodology was described as provided by the CRO. Size restrictions were carried out to align the findings from the SDS-Page where the 3-30 KDa being the prominent protein in the top hits.

  1. The preparation of the protein worries me a little. After ultra-filtration, I am unable to detect any high molecular weight proteins in the provided figures. I can see how additional processing could cause many or a few great candidates to degrade. Lowering the processing stages may increase the likelihood of maintaining protein integrity without causing protein degradation, in my opinion. In addition, I would like to know if the authors included protease inhibitors before preparing the sample. I cannot locate the protease inhibitor details in the manuscript. For instance, no proteins are present in lanes 4, 5, 9, and 10 in Figure 3 or in lanes 3, 8, and 13 in Figure 4. It's obvious that most of the proteins are being lost.

Reply: We acknowledge the reviewer’s concern. That was the limitation of our methodology. We did not use protease inhibitors. There was a possibility that some proteins could have been lost during the process. Regardless, our sample handling was uniform for all the materials which lead us pursue those with significant amount of proteins as hits. The aqueous extracts (Figure 3) and the C18 column fractions (Figure 4) were resuspended at equal weight/volume concentrations. The samples were not prepared specifically to preserve protein, as they had been processed by organic and aqueous extraction (Figure 3) and organic and aqueous extraction followed by a methanol gradient fractionation (Figure 4). The samples were only assessed for protein amounts to determine whether there were corresponding protein levels compared to activity levels observed in the PBMC cytotoxicity assay. The yields for each fraction in the C18 column fractionation (Figure 4) varied, but the concentration in the samples tested was equal. Finding proteins in every plant extract and C18 column fraction tested was unknown.

  1. To demonstrate that the proteins were similarly loaded onto the SDS-PAGE, I advise authors to do a western blot utilizing common housekeeping proteins. I've read over the sample preparation procedure. Authors have started with equal input raw material. However, because you could lose the proteins, processing stages like filtering, MW cutoff, etc., occasionally make a difference. Once more, sample preparation is crucial. I anticipate earning a point for using this approach. Again, the sample preparation matters a lot. I am expecting a valid point for using this method.

Reply: We appreciate the reviewers insight. We did not expect that similar protein amounts were loaded onto SDS-PAGE. As can be observed from the Coomassie blue stain, there were varying protein amounts in the samples tested. All samples were treated the same in their preparation, from the organic and aqueous extractions to the C18 column fractionations. We have a standard operating protocol for all extractions and fractionations that does not take protein preservation into consideration. The proteins that were observed were those that were present after multiple rounds of heating, drying, and treating with various solvents. It is possible and even likely that the proteins observed are denatured forms of higher molecular weight proteins, and it is possible that the original plants did not have much protein content to begin with.

  1. Data from LC-MS/MS must be submitted to the proteome consortium. 

Reply: It was under consideration. We will proceed as suggested.

Round 2

Reviewer 2 Report

Comments and Suggestions for Authors

I appreciate the authors' skillful response writing. I suggest the work be published in its existing form by the journal.